# Sample-Efficient Optimization in the Latent Space of Deep Generative Models via Weighted Retraining

**Austin Tripp**[*]
University of Cambridge
ajt212@cam.ac.uk

**Erik Daxberger**[*]
University of Cambridge
Max Planck Institute for
Intelligent Systems, Tübingen
ead54@cam.ac.uk

**José Miguel Hernández-Lobato**
University of Cambridge
Alan Turing Institute
Microsoft Research
jmh233@cam.ac.uk

## Abstract

Many important problems in science and engineering, such as drug design, involve optimizing an expensive black-box objective function over a complex, high-dimensional, and structured input space. Although machine learning techniques have shown promise in solving such problems, existing approaches substantially lack sample efficiency. We introduce an improved method for efficient black-box optimization, which performs the optimization in the low-dimensional, continuous latent manifold learned by a deep generative model. In contrast to previous approaches, we actively steer the generative model to maintain a latent manifold that is highly useful for efficiently optimizing the objective. We achieve this by periodically *retraining* the generative model on the data points queried along the optimization trajectory, as well as *weighting* those data points according to their objective function value. This weighted retraining can be easily implemented on top of existing methods, and is empirically shown to significantly improve their efficiency and performance on synthetic and real-world optimization problems.

## 1 Introduction

Many important problems in science and engineering can be formulated as optimizing an objective function over an input space. Solving such problems becomes particularly challenging when 1) the input space is high-dimensional and/or *structured* (i.e. discrete spaces, or non-Euclidean spaces such as graphs, sequences, and sets) and 2) the objective function is expensive to evaluate. Unfortunately, many real-world problems of practical interest have these characteristics. A notable example is drug design, which has a graph-structured input space, and is evaluated using expensive wet-lab experiments or time-consuming simulations. Recently, machine learning has shown promising results in many problems that can be framed as optimization, such as conditional image [68, 47] and text [51] generation, molecular and materials design [17, 58], and neural architecture search [16]. Despite these successes, using machine learning on structured input spaces and with limited data is still an open research area, making the use of machine learning infeasible for many practical applications.

One promising approach which tackles *both* challenges is a two-stage procedure that has emerged over the past few years, which we will refer to as *latent space optimization (LSO)* [20, 37, 41, 42, 48]. In the first stage, a (deep) generative model is trained to map tensors in a low-dimensional continuous space onto the data manifold in input space, effectively constructing a low-dimensional and continuous analog of the optimization problem. In the second stage, the objective function is optimized over this learned latent space using a surrogate model. Despite many successful applications in a variety of fields including chemical design [20, 28, 37, 9] and automatic machine learning [41, 42], LSO is primarily applied in a *post-hoc* manner using a pre-trained, general purpose generative model rather

---

[*]equal contribution

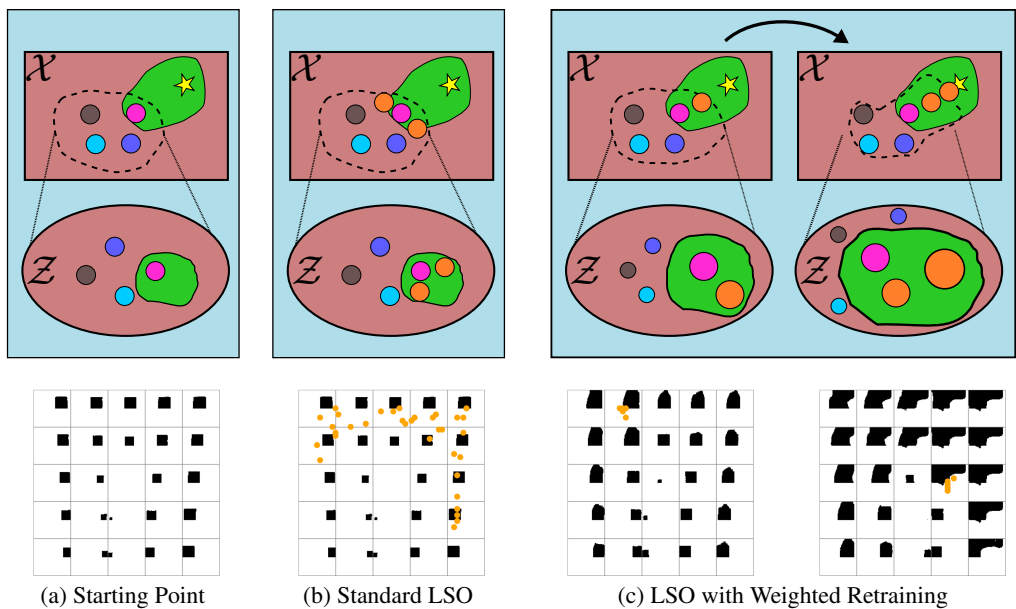

(a) Starting Point     (b) Standard LSO     (c) LSO with Weighted Retraining

Figure 1: Schematic illustrating LSO with and without weighted retraining. The cartoon illustrates the input/latent space of the generative model (**top**). The latent manifold from Section 6.2's 2D shape area maximization task is shown for comparison (**bottom**). Each image in the manifold shows the result of decoding a latent point on a uniform square grid in a 2D latent space; images are centered on the original grid points. Red/green regions correspond to points with low/high objective function values respectively. The yellow star is the global optimum in $\mathcal{X}$. Coloured circles are data points; their radius represents their weight. The dashed line surrounds the region of $\mathcal{X}$ modelled by $g$ (i.e. $g(\mathcal{Z})$, the image of $\mathcal{Z}$). **(a)** The status of the generative model $g$ at the start of optimization. **(b)** The result of standard LSO with $g$ fixed, which queries the points in orange. It is only able to find points close to the training data used to learn $\mathcal{Z}$, resulting in slow and incomplete exploration of $\mathcal{X}$. **(c)** The result midway (left) and at the end (right) of LSO with our proposed approach, which weights data points according to their objective function value and retrains $g$ to incorporate newly queried data. This continually adjusts $\mathcal{Z}$ to focus on modelling the most promising regions of $\mathcal{X}$, speeding up the optimization and allowing for substantial extrapolation beyond the initial training data.

than one trained specifically for the explicit purpose of downstream optimization. Put differently, the training of the generative model is effectively *decoupled* from the optimization task.

In this work, we identify and examine two types of decoupling in LSO. We argue that they make optimization unnecessarily difficult and fundamentally prevent LSO from finding solutions that lie far from the training data. Motivated by this, we propose *weighting of the data distribution* and *periodic retraining of the generative model* to effectively resolve this decoupling. We argue that these two modifications are highly complementary, fundamentally transforming LSO from a local optimizer into an efficient global optimizer capable of recursive self-improvement. Our contributions are:

1. We identify and describe two critical failure modes of previous LSO-based methods which severely limit their efficiency and performance, and thus practical applicability (Section 3).

2. We propose to combine dataset weighting with periodic retraining of the generative model used within LSO as an effective way to directly address the issued identified (Section 4).

3. We empirically demonstrate that weighted retraining significantly benefits LSO across a variety of application domains and generative models, achieving substantial improvements over state-of-the-art methods on a widely-used chemical design benchmark (Section 6).

## 2   Problem Statement and Background

**Sample-Efficient Black Box Optimization.** Let $\mathcal{X}$ be an *input space*, and let $f : \mathcal{X} \mapsto \mathbb{R}$ be an *objective function*. In particular, we focus on cases where 1) the input space $\mathcal{X}$ is *high-dimensional*

(i.e. 100+ effective dimensions) and *structured* (e.g. graphs, sequences or sets), and 2) the objective function $f(\mathbf{x})$ is *black-box* (i.e. no known analytic form or derivative information available) and is *expensive* to evaluate (e.g. in terms of time or energy cost). The *sample-efficient optimization* problem seeks to optimize $f$ over $\mathcal{X}$, evaluating $f$ as few times as possible, producing in the process a sequence of evaluated points $\mathcal{D}_M \equiv \{\mathbf{x}_i, f(\mathbf{x}_i)\}_{i=1}^{M}$ with $M$ function evaluations.

**Model-Based Optimization** seeks to approximate $f$ by constructing an *objective model* $h_\mathcal{X} : \mathcal{X} \mapsto \mathbb{R}$. $h_\mathcal{X}$ is then optimized as a surrogate for $f$. Although this is a widely used optimization technique, it can be difficult to apply in high-dimensional spaces (which may have many local minima) or in structured spaces (where any kind of discreteness precludes the use of gradient-based optimizers).

**Latent Space Optimization (LSO)** is a technique wherein model-based optimization is performed in the *latent space* $\mathcal{Z}$ of a *generative model* $g : \mathcal{Z} \mapsto \mathcal{X}$ that maps from $\mathcal{Z}$ to input space $\mathcal{X}$. To this end, a *latent objective model* $h : \mathcal{Z} \mapsto \mathbb{R}$ is constructed to approximate $f$ at the output of $g$, i.e. such that $f(g(\mathbf{z})) \approx h(\mathbf{z})$, $\forall \mathbf{z} \in \mathcal{Z}$. If $\mathcal{Z}$ is chosen to be a low-dimensional, continuous space such as $\mathbb{R}^n$, the aforementioned difficulties with model-based optimization can be avoided, *effectively turning a discrete optimization problem into a continuous one*. To realize LSO, $g$ can be chosen to be a state-of-the-art deep generative model (DGM), such as a variational autoencoder (VAE) [35, 55] or a generative adversarial network (GAN) [21], which have been shown to be capable of learning vector representations of many types of high-dimensional, structured data [5, 13, 63, 70]. Furthermore, $h$ can be chosen to be a flexible probabilistic model such as a Gaussian process [73], allowing sample-efficient *Bayesian optimization* to be performed [6, 61]. $h$ can be trained by using an approximate inverse to $g$, $q : \mathcal{X} \mapsto \mathcal{Z}$, to find a corresponding latent point $\mathbf{z}_i$ for each data point $\mathbf{x}_i$.

# 3 Failure Modes of Latent Space Optimization

To understand the shortcomings of LSO, it is necessary to first examine in detail the role of the generative model, which is usually a DGM. State-of-the-art DGMs such as VAEs and GANs are trained with a prior $p(\mathbf{z})$ over the latent space $\mathcal{Z}$. This means that although the resulting function $g : \mathcal{Z} \mapsto \mathcal{X}$ is defined over the entire latent space $\mathcal{Z}$, it is effectively only trained on points in regions of $\mathcal{Z}$ with high probability under $p$. Importantly, even if $\mathcal{Z}$ is an unbounded space with infinite volume such as $\mathbb{R}^n$, because $p$ has finite volume, there must exist a *finite* subset $\mathcal{Z}' \subset \mathcal{Z}$ that contains virtually all the probability mass of $p$. We call $\mathcal{Z}'$ the *feasible region* of $\mathcal{Z}$. Although in principle optimization can be performed over all of $\mathcal{Z}$, it has been widely observed that optimizing outside of the feasible region tends to give poor results, yielding samples that are low-quality, or even invalid (e.g. invalid molecular strings, non-grammatical sentences); therefore all LSO methods known to us employ some sort of measure to restrict the optimization to near or within the feasible region [20, 37, 48, 22, 72, 43, 10]. This means that LSO should be treated as a *bounded* optimization problem, whose feasible region is determined by $p$.

Informally, the training objective of $g$ encourages points sampled from within the feasible region to match the data distribution that $g$ was trained on, effectively "filling" the feasible region with points similar to the dataset, such that a point's relative volume is roughly proportional to its frequency in the training data. For many optimization problems, most of the training data for the DGM is low-scoring (i.e. highly sub-optimal objective function values), thereby causing most of the feasible region to contain low-scoring points. Not only does this make the optimization problem more difficult to solve (like finding the proverbial "needle in a haystack"), but actually leaves insufficient space in the feasible region for a large number of novel, high-scoring points that lie outside the training distribution to be modelled by the DGM. Therefore, even a perfect optimization algorithm with unlimited evaluations of the objective function might be unable to find a novel point that is substantially better than the best point in the original dataset, simply because such a point may not exist in the feasible region.

This pathological behaviour is conceptually illustrated in Figure 1b, where LSO is unable to find or even approach the global optimum that lies far from the training data. We propose that LSO's performance is severely limited by two concrete problems in its setup. The first problem is that the generative model's training objective (to learn a latent space that captures the data distribution as closely as possible), does not necessarily match the true objective (to learn a latent space that is amenable to efficient optimization of the objective function). Put in terms of the cartoon in Figure 1b, the feasible region that is learned, which uniformly and evenly surrounds the data points, is not the feasible region that would be useful for optimization, which would model more of the green region at

the expense of the red region. This is also seen in the 2D shape area maximization task in Figure 1b, where the latent manifold contains only low-area shapes that the model was trained on, and nothing close to the all-black global optimum. The second problem is that information on new points acquired during the iterative optimization procedure is not propagated to the generative model, where it could potentially help to refine and expand the coverage of the feasible region, uncovering new promising regions that an optimization algorithm can exploit. In terms of Figure 1b, the new data is not used to shift the feasible region toward the green region, despite the optimization process indicating that this is a very promising region of $\mathcal{X}$ for optimization. Luckily, we believe that neither of these two problems is inherent to LSO, and now pose a framework that directly addresses them.

## 4   Latent Space Optimization with Weighted Retraining

### 4.1   Training a Generative Model with a Weighted Training Objective

While it is unclear in general how to design a generative model that is maximally amenable to LSO, the argument presented in Section 3 suggests that it would at least be beneficial to dedicate a higher fraction of the feasible region to modelling high-scoring points. One obvious but inadequate method of achieving this is to simply discard all low-scoring points from the dataset used to train the DGM, e.g. by keeping only the top 10% of the data set (in terms of score). While this strategy could be feasible if data is plentiful, when data is scarce this option may not be viable because state-of-the-art neural networks need a large amount of training data to avoid overfitting. This issue can be resolved by not viewing inclusion in the dataset as a binary choice, but instead as a *continuum* that can be realized by *weighting* the data points unevenly. If the generative model is trained on a distribution that systematically places more probability mass on high-scoring points and less mass on slow scoring points, the distribution-matching term in the DGM's training objective will incentivize a larger fraction of the feasible region's volume to be used to model high-scoring points, while simultaneously using all known data points to learn useful representations and avoid overfitting.

A simple way to achieve this weighting is to assign an explicit weight $w_i$ to each data point, such that $\sum_i w_i = 1$. As the training objective of common DGMs involves the expected value of a loss function $\mathcal{L}$ with respect to the data distribution,[2] weighted training can be implemented by simply replacing the empirical mean over the training data with a *weighted* empirical mean: i.e. $\sum_{\mathbf{x}_i \in \mathcal{D}} w_i \mathcal{L}(\mathbf{x}_i)$ instead of $\sum_{\mathbf{x}_i \in \mathcal{D}} \frac{1}{N} \mathcal{L}(\mathbf{x}_i)$. In practice, mini-batch stochastic gradient descent is used to optimize this objective to avoid summing over all data points. Unbiased mini-batches can be constructed by sampling each data point $\mathbf{x}_i$ with probability $w_i$ with replacement to construct each batch (see Appendix A.2 for more details).

We offer no universal rules for setting weights, except that all weights $w_i$ should be restricted to strictly positive values, because a negative weight would incentivize the model to perform poorly, and a weight of zero is equivalent to discarding a point. This aside, there are many reasonable ways to choose the weights such that high-scoring points are weighted more, and low-scoring points are weighted less. In this work, we decide to use a rank-based weight function,

$$w(\mathbf{x}; \mathcal{D}, k) \propto \frac{1}{kN + \mathrm{rank}_{f,\mathcal{D}}(\mathbf{x})}, \quad \mathrm{rank}_{f,\mathcal{D}}(\mathbf{x}) = |\{\mathbf{x}_i : f(\mathbf{x}_i) > f(\mathbf{x}), \ \mathbf{x}_i \in \mathcal{D}\}| \ , \quad (1)$$

which assigns a weight roughly proportional to the reciprocal (zero-based) rank of each data point. We chose Equation (1) because it yields weights which are always positive, resilient to outliers, and has stable behaviour over a range of dataset sizes (this is explained further in Appendix A.1). Furthermore, as shown in Figure 2, it admits a single tunable hyperparameter $k$ which continuously controls the degree of weighting, where $k = \infty$ corresponds to uniform weighting, i.e. $w_i = \frac{1}{N}, \forall i$, while $k = 0$ places *all* mass on only the single point with the highest objective function value.

### 4.2   Periodic Retraining to Update the Latent Space

To allow the latent manifold to adapt to new information, we propose a conceptually simple solution: *periodically retraining the generative model* during the optimization procedure. In practice, this could be done by training a new model from scratch, or by fine-tuning the previously trained model on the novel data. However, as is often pointed out in the active learning literature, the effect of adding a few

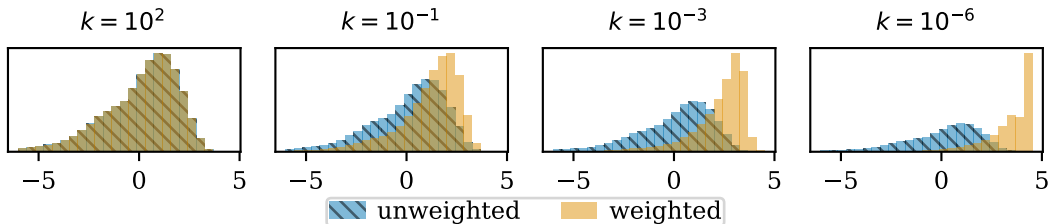

Figure 2: Histogram of objective function values for the ZINC dataset (see Section 6) with uniform weighting (in blue) as well as rank weighting from Equation (1) for different $k$ values (in orange). Large $k$ approaches uniform weighting, while small $k$ places most weight on high-scoring points.

---

**Algorithm 1** Latent Space Optimization with Weighted Retraining (changes highlighted in blue)

1: **Input:** Data $\mathcal{D} = \{(\mathbf{x}_i, f(\mathbf{x}_i))\}_{i=1}^{N}$, query budget $M$, objective function $f(\mathbf{x})$, latent objective model $h(\mathbf{z})$, generative/inverse model $g(\mathbf{z})/q(\mathbf{x})$, retrain frequency $r$, weighting function $w(\mathbf{x})$
2: **for** $1, \ldots, M/r$ **do**
3:     Train generative model $g$ and inverse model $q$ on data $\mathcal{D}$ weighted by $\mathcal{W} = \{w(\mathbf{x})\}_{\mathbf{x} \in \mathcal{D}}$
4:     **for** $1, \ldots, r$ **do**
5:         Compute latent variables $\mathcal{Z} = \{\mathbf{z} = q(\mathbf{x})\}_{\mathbf{x} \in \mathcal{D}}$
6:         Fit objective model $h$ to $\mathcal{Z}$ and $\mathcal{D}$, and optimize $h$ to obtain new latent query point $\tilde{\mathbf{z}}$
7:         Obtain corresponding input $\tilde{\mathbf{x}} = g(\tilde{\mathbf{z}})$, evaluate $f(\tilde{\mathbf{x}})$ and set $\mathcal{D} \leftarrow \mathcal{D} \cup \{(\tilde{\mathbf{x}}, f(\tilde{\mathbf{x}}))\}$
8:     **end for**
9: **end for**
10: **Output:** Augmented dataset $\mathcal{D}$

---

additional points to a large dataset is rather negligible, and thus it is unlikely that the generative model will change significantly if retrained on this augmented dataset [60]. While one could also retrain on *only* the new data, this might lead to the well-known phenomenon of catastrophic forgetting [44].

A key observation we make is that *the data weighting outlined in Section 4.1 actually resolves this problem*. Specifically, if the new points queried are high-scoring, then a suitable weighting scheme (such as Equation (1)) will assign a large weight to them, while simultaneously decreasing the weights of many of the original data points, meaning that a small number of new points can have a disproportionate impact on the training distribution. If the generative model is then retrained using this distribution, it can be expected to change significantly to incorporate these new points into the the latent space in order to minimize the weighted loss. By contrast, if the new points queried are low-scoring, then the distribution will change negligibly, and the generative model will not significantly update, thereby avoiding adding new low-scoring points into the feasible region.

### 4.3 Weighted Retraining Combined

When put together, *data weighting* and *periodic retraining* complement each other elegantly, transforming the generative model from a passive decoding function into an active participant in the optimization process, whose role is to ensure that the latent manifold is constantly occupied by the most updated and relevant points for optimization. Their combined effect is visualized conceptually in Figure 1c. In the first iteration, weighted training creates a latent space with more high scoring points, causing the feasible region to extend farther into the green region at the expense of the red region. This allows a better orange point to be chosen relative to Figure 1b. In the second iteration in Figure 1c, weighted training with the orange point incorporates even more high-scoring points into the latent space, allowing an even better point to be found. Qualitatively similar results can be seen in the 2D shape area maximization task, where weighted retraining introduces points with very high areas into the latent space compared to Figure 1a (details for this experiment are given in Section 6).

In the remainder of this paper, we refer to the combination of these techniques as *weighted retraining* for brevity; see Algorithm 1 for pseudocode. We highlight that this algorithm is straightforward to implement in most models, with brief examples given in Appendix A.3. Computationally, the overhead of the weighting is minimal, and the cost of the retraining can be reduced by fine-tuning

an existing model on the weighted dataset instead of retraining it from scratch. Although this may still be prohibitively expensive for some applications, we stress that in many scenarios the cost of training a model is insignificant compared to even a single evaluation of the objective function (e.g. performing wet-lab experiments for drug design), making weighted retraining a sensible choice.

## 5 Related Work

While a large body of work is applicable to the general problem formulated in Section 2 (both with and without machine learning), below we focus only on the most relevant machine learning literature.

**Latent Space Optimization.** Early formulations of LSO were motivated by scaling Gaussian processes (GPs) to high dimensional problems with simple linear manifolds, using either random projections [71] or a learned transformation matrix [19]. LSO using DGMs was first applied to chemical design [20], and further built upon subsequently [28, 37, 15, 31, 9, 10, 22, 43]. It has also been applied to other fields, e.g. automatic machine learning [41, 42, 76], conditional image generation [48, 47], and model explainability [2]. If the surrogate model is a GP, the DGM can be viewed as an "extended kernel", making LSO conceptually related to deep kernel learning [74, 26].

**Weighted Retraining.** A few previous machine learning methods can be viewed as implementing a version of weighted retraining. The cross-entropy (CE) method iteratively retrains a generative model using a weighted training set, such that high-scoring points receive higher weights [57, 56, 12]. Indeed, particular instantiations of the CE method such as reward-weighted regression [53], feedback GAN [24], and design/conditioning by adaptive sampling (DbAS/CbAS) [8, 7] have been applied to similar problem settings as our work. However, our proposed method of weighted retraining has two main differences from CE. Firstly, *standard CE produces only binary weights* [12], which amounts to simply adding or removing points from the training set.[3] This is sub-optimal for reasons discussed in Sections 4.1 and 4.2, and consequently, *we consider a strictly more general form of weighting*. Secondly, *CE has no intrinsic optimization component.* High-performing points are found only by repeatedly sampling from the generative model and evaluating $f$. By contrast, our method *explicitly selects high-performing points* using Bayesian optimization. The necessity of repeated sampling in CE makes it only suitable in cases where evaluating $f$ is cheap, which is *not* what we are considering. Moreover, works such as [59] perform optimization by fine-tuning a generative model on a smaller dataset of high-scoring samples. This can also be viewed as a special case of weighted retraining with binary weights, where the weights are implicitly defined by the number of fine-tuning epochs.

**Bayesian optimization (BO)** is a technique that maintains a probabilistic model of the objective function, and chooses new points to evaluate based on the modelled distribution of the objective value at unobserved points. BO is widely viewed as the go-to framework for sample-efficient black-box optimization [6, 64]. However, most practical BO models exist for continuous, low-dimensional spaces [61]. Recent works have tried to develop models to extend BO to either structured [3, 34, 11, 49] or high-dimensional [32, 46, 25] input spaces. To our knowledge, only BO methods with a significant amount of domain-specific knowledge infused into their design are able to handle input spaces that are *both* high-dimensional and structured. A noteworthy example is ChemBO which uses both a customized molecular kernel and a synthesis graph to perform BO on molecules [36], which we compare against in Section 6. In contrast, our method can be applied to any problem without domain knowledge, and has comparable performance to ChemBO. Finally, in an interesting parallel to our work, [4] use a weighted acquisition function to increase the sample efficiency of BO.

**Reinforcement learning (RL)** frames optimization problems as Markov decision processes for which an agent learns an optimal policy [66]. It has recently been applied to various optimization problems in structured input spaces [38], notably in chemical design [75, 77, 23, 50, 54, 62]. While RL is undoubtedly effective at optimization, it is generally extremely sample inefficient, and consequently its biggest successes are in virtual environments where function evaluations are inexpensive [38].

**Conditional Generative Models.** Finally, one interesting direction is the development of *conditional generative models*, which directly produce novel points conditioned on a specific property value [65, 45]. Although many variants of these algorithms have been applied to real-world problems such as chemical design [29, 33, 39, 40, 8], the sample efficiency of this paradigm is currently unclear.

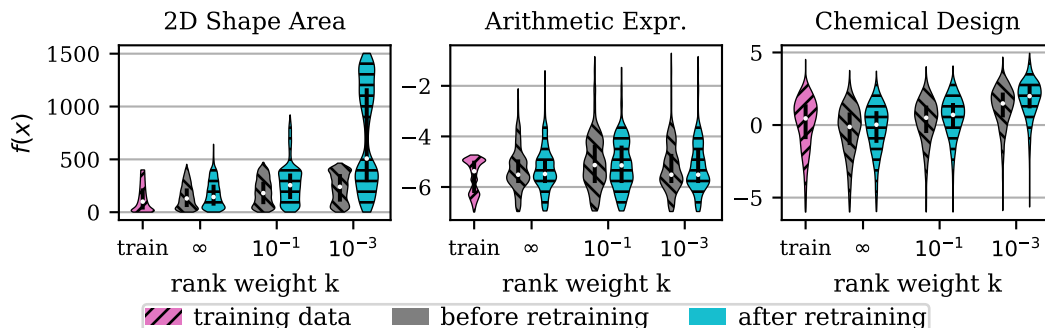

Figure 3: Objective value distribution for the training set and samples from the DGM's prior for all three tasks for different $k$ values, before and after weighted retraining (see Section 6.2).

## 6 Empirical Evaluation

This section aims to empirically answer three main questions:

1. How does weighted training affect the latent space of DGMs? (Section 6.1)
2. How do the parameters of weighted retraining influence optimization? (Section 6.2)
3. Does weighted retraining compare favourably to existing methods? (Section 6.3)

To answer these questions, we perform experiments using three optimization tasks chosen to represent three different data and model types. The tasks are described in more detail below. Because there is no obvious single metric to evaluate sample-efficient optimization, we choose to plot the $K$th best novel evaluated point as a function of the number of objective function evaluations, which we denote as the *TopK* score (details in Appendix C.3). All plots show the average performance and standard deviation across runs with 5 different random seeds unless otherwise stated. This evaluation method is common practice in Bayesian optimization [61]. It contrasts with previous works which typically report only final scores, and take the maximum across seeds rather than the average [20, 37, 9, 28].

**2D Shape Area Maximization Toy Task.** As a simple toy task that can be easily visualized in 2D, we optimize for the shape with the largest total area in the space of $64 \times 64$ binary images (i.e. the largest number of pixels with value 1). **Data:** A dataset of $\approx$10,000 squares of different sizes and positions on a $64 \times 64$ background, with a maximum area of 400 (see Figure 14 in Appendix C for examples). **Model:** A convolutional VAE with $\mathcal{Z} = \mathbb{R}^2$, as a standard neural network architecture for image modelling. **Latent Optimizer:** We enumerate a grid in latent space over $[-3, +3]^2$, to emulate a perfect optimizer for illustration purposes (this is only feasible since $\mathcal{Z}$ is low-dimensional).

**Arithmetic Expression Fitting Task.** We follow [37] and optimize in the space of single-variable arithmetic expressions generated by a formal grammar. Examples of such expressions are `sin(2)`, `v/(3+1)` and `v/2 * exp(v)/sin(2*v)`, which are all considered to be functions of some variable `v`. Following [37], the objective is to find an expression with minimal mean squared error to the target expression $x^* = $ `1/3 * v * sin(v*v)`, computed over 1000 values of `v` evenly-spaced between $-10$ and $+10$. **Data:** 50,000 univariate arithmetic expressions generated by the formal grammar from [37]. **Model:** A grammar VAE [37], chosen because of its ability to produce only valid grammatical expressions. **Latent Optimizer:** Bayesian optimization with the expected improvement acquisition function [30] and a sparse Gaussian process model with 500 inducing points [67], following [37].

**Chemical Design Task.** We follow [20] and optimize the drug properties of molecules. In particular, we consider the standardized task originally proposed in [20] of synthesizing a molecule with maximal penalized *water-octanol partition coefficient* (logP), starting from the molecules in the ZINC250k molecule dataset [27] (see Appendix C.6 for more details). This task has been studied in a long series of papers performing optimization in chemical space, allowing the effect of weighted retraining to be quantitatively compared to other optimization approaches [37, 9, 28, 77, 75]. **Data:** The ZINC250k molecule dataset [27], using the same train/test split as [28]. **Model:** A junction tree VAE [28], chosen because it is a state-of-the-art VAE for producing valid chemical structures. For direct comparability to previous results, we use the pre-trained model provided in the code repository of

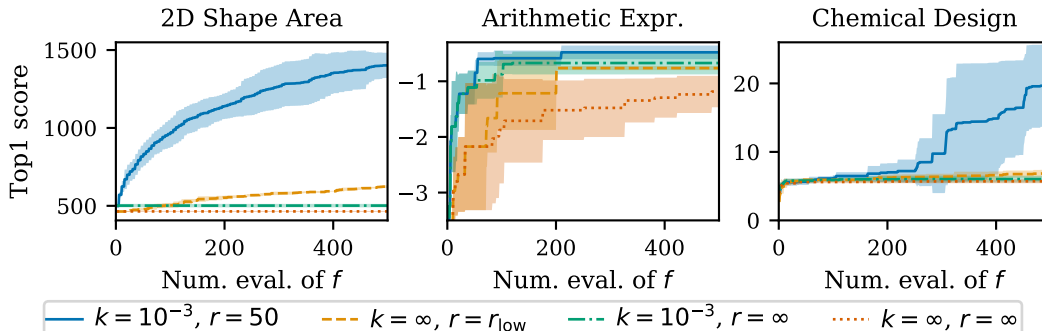

Figure 4: Top1 optimization performance of weighted retraining for all tasks, for different $k$ values (i.e. $k \in \{10^{-3}, \infty\}$) and retraining frequencies (i.e. $r_{\text{low}} = 5$ for the 2D shape area task, and $r_{\text{low}} = 50$ for the other two tasks). Shaded area corresponds to standard deviation.

[28] as the unweighted model, and create weighted models by fine-tuning the pre-trained model for 1 epoch over the full weighted dataset. **Latent Optimizer:** Same as for the arithmetic expression task.

More details on the experimental setup are given in Appendix C. All experimental data and code to reproduce the experiments can be found at `https://github.com/cambridge-mlg/weighted-retraining`.

## 6.1 Effect of Weighted Training

In this section, we seek to validate some of the conjectures made in Sections 3 and 4, namely that 1) the latent space of a DGM trained on uniformly weighted data contains many poor-performing points, and 2) that weighted training fixes this by introducing more high-performing points into the latent space. To test this, we train a VAE for each task using rank weighting with a variety of $k$ values (noting that $k = \infty$ corresponds to uniform weighting), initializing the weights using a pre-trained VAE to ensure that the different runs are comparable. We evaluate $f$ on samples from the DGM's prior for each task, and plot the resulting distributions in Figure 3 with the label *before retraining*. Although the distribution of scores for $k = \infty$ does not exactly match the training distribution for any example, it tends to have a similar range, showing that much of the latent space is dedicated to modelling low-scoring points. Weighted training robustly causes the distribution to skew towards higher values at the expense of lower values, which is exactly the intended effect. The upshot is that the result on all 3 tasks broadly supports our conjectures.

## 6.2 Effect of Weighted Retraining Parameters on Optimization

When using rank-weighting from Equation (1) with parameter $k$ and picking a fixed period for model retraining $r$, LSO with weighted retraining can be completely characterized by $k$ and $r$. The baseline of uniform weighting and no retraining is represented by $k = r = \infty$, with decreasing values of $k$ and $r$ representing more skewed weighting and more frequent retraining, respectively. For each task, we choose a value $r_{\text{low}}$ based on our computational retraining budget, then perform LSO for each value of $k \in \{k_{\text{low}}, \infty\}$ and $r \in \{r_{\text{low}}, \infty\}$. For computational efficiency retraining is done via fine-tuning. Further experimental details are given in Appendix C.

The results are shown in Figure 4. Firstly, comparing the case of $k = \infty, r = \infty$ with $k = \infty, r = r_{\text{low}}$ and $k = k_{\text{low}}, r = \infty$ suggests that both weighting and retraining help individually, as hypothesized in Section 4. Secondly, in all cases, weighted retraining with $k = k_{\text{low}}, r = r_{\text{low}}$ performs better than all other methods, suggesting that they have a synergistic effect when combined. Note that the performance often increases suddenly after retraining, suggesting that the retraining does indeed incorporate new information into the latent space, as conjectured. Lastly, the objective function values of prior samples from the models after weighted retraining with $r = r_{\text{low}}$ is shown in Figure 3 in blue. In all cases, the distribution becomes more skewed towards positive values, with the difference being more pronounced for lower $k$ values. This suggests that weighted retraining is able to significantly modify the latent space, even past the initial retraining. See Appendix B for results with a larger set of $k$ and $r$ values, and Top$K$ plots for other values of $K$.

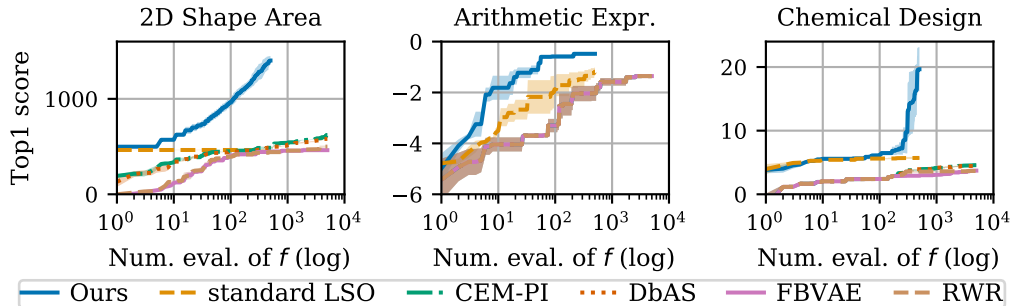

Figure 5: Comparison of weighted retraining, LSO, CEM-PI, DbAS, FBVAE and RWR. Our approach significantly outperforms all baselines, achieving both better sample-efficiency and final performance.

## 6.3 Comparison with Other Methods

Finally, we compare our proposed method of LSO with weighted retraining with other methods on the same tasks. The first class of methods are based on the cross-entropy method as discussed in Section 5, namely design by adaptive sampling (DbAS) [8], the cross-entropy method with probability of improvement (CEM-PI) [56], the feedback VAE (FBVAE) [24] and reward-weighted regression (RWR) [53]. These methods are noteworthy because they can be viewed as a particular case of weighted retraining, where the weights are binary (except for DbAS) and the latent optimizer simply consists of sampling from the DGM's prior. The hyperparameters of these methods are the sequence of quantiles, and the retraining frequency. We optimize these hyperparameters using a grid search, as detailed in Appendix C. Figure 5 shows the performance of these methods on the best hyperparameter setting found, as a function of the number of samples drawn (with a budget of 5,000 samples in total). We plot the average and standard deviation across 3 random seeds, as we found the variances to be relatively low. We observe that all other forms of weighted retraining perform significantly worse than our own, failing to achieve the performance of our approach, even with an evaluation budget that is an order of magnitude larger than ours (i.e. 5,000 vs 500). We attribute this both to their binary weighting scheme and their lack of a sample-efficient latent optimizer.

Secondly, we compare against other methods in the literature that have attempted the same chemical design task. To our knowledge, the best previously reported score obtained using a machine learning method is 11.84 and was obtained with $\approx 5000$ samples [77]. By contrast, our best score is 27.84 and was achieved with only 500 samples. Expanding the scope to include more domain-specific optimization methods, we acknowledge that ChemBO achieved an impressive score of 18.39 in only 100 samples [36], which is better than our method's performance with only 100 samples. Table 1 in the appendix gives a more detailed comparison with other work.

## 7 Discussion and Conclusion

We proposed a method for efficient black-box optimization over high-dimensional, structured input spaces, combining latent space optimization with weighted retraining. We showed that while being conceptually simple and easy to implement on top of previous methods, weighted retraining significantly boosts their efficiency and performance on challenging real-world optimization problems.

There are several drawbacks to our method that are promising directions for future work. Firstly, we often found it difficult to train a latent objective model that performed well at optimization in the latent space, which is critical for good performance. We believe that further research is necessary into other techniques to make the latent space of DGMs more amenable to optimization. Secondly, our method requires a large dataset of labelled data to train a DGM, making it unsuitable for problems with very little data, motivating adaptations that would allow unlabelled data to be utilized. Finally, due to being relatively computationally intensive, we were unable to characterize the long-term optimization behaviour of our algorithm. In this regime, we suspect that it may be beneficial to use a weighting *schedule* instead of a fixed weight, which may allow balancing exploration vs. exploitation similar to simulated annealing [69]. Overall, we are excited at the potential results of LSO and hope that it can be applied to a variety of real-world problems in the near future.

## Broader Impact

Ultimately, this work is preliminary, despite the promise of latent space optimization, there may still be significant obstacles to applying it more widely in the real world. That aside, we believe that the primary effect of this line of research will be to enable faster discoveries of novel entities, such as new medicines, new energy materials, or new device designs. The worldwide effort to develop vaccines and treatments for the COVID-19 pandemic has highlighted the importance of techniques for fast, targeted discovery using only small amounts of data: we have seen that even if the whole world is devoted to performing experiments with a single goal, the sheer size of the search space means that sample efficiency is still important.

As much as this technology could be used to discover good things, it could also be used to discover bad things (e.g. chemical/biological weapons). However, as substantial resources and infrastructure are required to produce these, we do not expect that this line of work will enable new parties to begin their development. Rather, at worse, it may allow people who are already involved in their development to do it slightly more effectively.

Finally, we believe that this line of work has the potential to influence other problem areas in machine learning, such as conditional image generation and conditional text generation, because these tasks can also be viewed as optimization, whose objective function is a human judgement, which is generally expensive to obtain.

## Acknowledgments and Disclosure of Funding

We thank Ross Clarke, Gregor Simm, David Burt, and Javier Antorán for insightful feedback and discussions. AT acknowledges funding via a C T Taylor Cambridge International Scholarship. ED acknowledges funding from the EPSRC and Qualcomm. This work has been performed using resources provided by the Cambridge Tier-2 system operated by the University of Cambridge Research Computing Service (http://www.hpc.cam.ac.uk) funded by EPSRC Tier-2 capital grant EP/P020259/1.

## Footnotes

[2]For a VAE, $\mathcal{L}$ is the per-datapoint ELBO [35], while for a GAN, $\mathcal{L}$ is the discriminator score [21].

[3]Although methods such as DbAS [8] generalize these weights to lie in $[0, 1]$, this is determined by the noise of the oracle and therefore will still produce binary weights when $f$ is deterministic, as considered in this paper.

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
