[Supplementary Material]

# A  Details on the Weighting Function

## A.1  More Information on Rank-Based Weighting

**Independence from Dataset Size**  We show that the key properties of rank-based weighting depend *only* on $k$, and not on the dataset size $N$, meaning that applying rank weighting with a fixed $k$ to differently sized datasets will yield similar results. In particular, we show that under mild assumptions, the fraction of weights devoted to a particular quantile of the data depends on $k$ but not $N$.

Suppose that the quantile of interest is the range $q_1$–$q_2$ (for example, the first quartile corresponds to the range 0–0.25). This corresponds approximately to the points with ranks $q_1 N$–$q_2 N$. We make the following assumptions:

1. $kN \gg 1$
2. $kN$ is approximately integer valued, which is realistic if $N \gg 1/k$
3. $q_1$ and $q_2$ are chosen so that $q_1 N$ and $q_2 N$ are integers.

Because the ranks form the sequence $0, 1, \ldots, N-1$, under the above assumptions all weights are reciprocal integers, so the sum of the rank weights is strongly connected to the harmonic series. Recall that the partial sum of the harmonic series can be approximated by the natural logarithm:

$$\sum_{j=1}^{N} \frac{1}{j} \approx \ln N + \gamma \tag{2}$$

Here, $\gamma$ is the Euler–Mascheroni constant. The fraction of the total weight devoted to the quantile $q_1$–$q_2$ can be found by summing the weights of points with rank $q_1 N$–$q_2 N$, and dividing by the normalization constant (the sum of all weights). First, because $kN \gg 1$ implies that $(kN-1) \approx kN$, the sum of all the weights can be expressed as:

$$\begin{aligned}
\sum_{r=0}^{N-1} w(\mathbf{x}_r; \mathcal{D}, k) &= \sum_{r=0}^{N-1} \frac{1}{kN+r} \\
&= \sum_{r=1}^{kN+(N-1)} \frac{1}{r} - \sum_{r'=1}^{kN-1} \frac{1}{r'} \\
&\approx (\ln\left((k+1)N-1\right) + \gamma) - (\ln\left(kN-1\right) + \gamma) \\
&= \ln \frac{(k+1)N-1}{kN-1} \approx \ln \frac{(k+1)N}{kN} = \ln\left(1 + \frac{1}{k}\right)
\end{aligned}$$

Note that this does not depend on the dataset size $N$. Second, using the same assumption, the sum of the weights in the quantile is:

$$\begin{aligned}
\sum_{r=q_1 N}^{q_2 N} w(\mathbf{x}_r; \mathcal{D}, k) &= \sum_{r=q_1 N}^{q_2 N} \frac{1}{kN+r} \\
&= \sum_{r=1}^{(k+q_2)N} \frac{1}{r} - \sum_{r'=1}^{(k+q_1)N-1} \frac{1}{r'} \\
&\approx (\ln\left((k+q_2)N-1\right) + \gamma) - (\ln\left((k+q_1)N-1\right) + \gamma) \\
&= \ln \frac{(k+q_2)N}{(k+q_1)N-1} \approx \ln \frac{(k+q_2)N}{(k+q_1)N} = \ln \frac{(k+q_2)}{(k+q_1)}
\end{aligned}$$

which is also independent of $N$ (note that setting $q_1 = 0$, $q_2 = 1$ into the formula yields the same expression for the sum of the weights as derived above). Therefore, the fraction of the total weight allocated to a given quantile of data is *independent of $N$*, being only dependent on $k$. Although the analysis that led to this result made some assumptions about certain values being integers, in practice

Figure 6: Cumulative distribution of rank weights (sorted highest to lowest), showing a distribution that is independent of $N$ if $kN > 1$.

the actual distributions of weights are extremely close to what this analysis predicts. Figure 6 shows the allocation of the weights to different quantiles of the datasets. For $kN > 1$, the distribution is essentially completely independent of $N$. Only when $kN < 1$ this fails to hold.

Finally, we discuss some potential questions about the rank-based weighting.

**Why do the weights need to be normalized?** If the objective is to minimize $\sum_{\mathbf{x}_i \in \mathcal{D}} w_i \mathcal{L}(\mathbf{x}_i)$, for any $a > 0$, minimizing $a \sum_{\mathbf{x}_i \in \mathcal{D}} w_i \mathcal{L}(\mathbf{x}_i)$ is an equivalent problem. Therefore, in principle, the absolute scale of the weights does not matter, and so the weights do not *need* to be normalized, even if this precludes their interpretation as a probability distribution. However, in practice, if minimization is performed using gradient-based algorithms, then the scaling factor for the weights is also applied to the gradients, possibly requiring different hyperparameter settings (such as a different learning rate). By normalizing the weights, it is easier to identify hyperparameter settings that work robustly across different problems, thereby allowing weighted retraining to be applied with minimal tuning.

**Why not use a weight function directly based on the objective function value?** Although there is nothing inherently flawed about using such a weight function, there are some practical difficulties.

- Such a weight function would either be bounded (in which case values beyond a certain threshold would all be weighted equally), or it would be very sensitive to outliers (i.e. extremely high or low values which would directly cause the weight function to take on an extremely high or low value). This is extremely important because the weights are *normalized*, so one outlier would also affect the values of all other points.

- Such a weight function would not be invariant to simple transformations of the objective function. For example, if the objective function is $f$, then maximizing $f(\mathbf{x})$ or $f_{ab}(\mathbf{x}) = a f(\mathbf{x}) + b$ is an equivalent problem (for $a > 0$), but would yield different weights. This would effectively introduce scale hyperparameters into the weight function, which is undesirable.

## A.2 Mini-Batching for Weighted Training

As mentioned in the main text, one method of implementing the weighting with mini-batch stochastic gradient descent is to sample each point $x_i$ with probability proportional to its weight $w_i$ (with replacement). A second method is to sample points with uniform probability and re-weight each point's contribution to the total loss by its weight:

$$\sum_{\mathbf{x}_i \in \mathcal{D}} w_i \mathcal{L}(\mathbf{x}_i) \approx \frac{N}{n} \sum_{j=1}^{n} w_j \mathcal{L}(\mathbf{x}_j) \tag{3}$$

If done naively, these mini-batches may have extremely high variance, especially if the variance of the weights is large. In practice, we found it was sufficient to reduce the variance of the weights by simply adding multiple copies of any $\mathbf{x}_i$ with $w_i > w_{\max}$, then reducing the weight of each copy such that the sum is still $w_i$. The following is a `Python` code snippet implementing this variance reduction:

```python
def reduce_variance(data, weights, w_max):
    new_data = []
    new_weights = []
    for x, w in zip(data, weights):
        if w <= w_max:  # If it is less than the max weight, just add it
            new_data.append(x)
            new_weights.append(w)
        else:  # Otherwise, add multiple copies
            n_copies = int(math.ceil(w / w_max))
            new_data += [x] * n_copies
            new_weights += [w / n_copies] * n_copies
    return new_data, new_weights
```

The parameter `w_max` was typically set to $5.0$, which was chosen to both reduce the variance, while simultaneously not increasing the dataset size too much. Note that this was applied *after* the weights were normalized. This also makes it feasible to train for only a fraction of an epoch, since without variance reduction techniques there is a strong possibility that high-weight data points would be missed if the entire training epoch was not completed.

### A.3 Implementation of Weighted Training

One of the benefits of weighted retraining which we would like to highlight is its ease of implementation. Below, we give example implementations using common machine learning libraries.

#### A.3.1 PyTorch (weighted sampling)

**Standard Training**

```python
from torch.utils.data import *

dataloader = DataLoader(data)
for batch in dataloader:
    # ...
```

**Weighted Training**

```python
from torch.utils.data import *
sampler = WeightedRandomSampler(
    weights, len(data))
dataloader = DataLoader(data, sampler=sampler)
for batch in dataloader:
    # ...
```

#### A.3.2 PyTorch (direct application of weights)

**Standard Training**

```python
criterion = nn.MSELoss()
outputs = model(inputs)
loss = criterion(outputs, targets)

loss.backward()
```

**Weighted Training**

```python
criterion = nn.MSELoss(reduction=None)
outputs = model(inputs)
loss = criterion(outputs, targets)
loss = torch.mean(loss * weights)
loss.backward()
```

#### A.3.3 Keras

**Standard Training**

```python
model.fit(x, y)
```

**Weighted Training**

```python
model.fit(x, y, sample_weight=weights)
```

### A.4 Implementation of Rank Weighting

We provide a simple implementation of rank-weighting:

```python
import numpy as np
def get_rank_weights(outputs, k):

    # argsort argsort to get ranks (a cool trick!)
    # assume here higher outputs are better
    outputs_argsort = np.argsort(-np.asarray(outputs))
    ranks = np.argsort(outputs_argsort)
    return 1 / (k * len(outputs) + ranks)
```

## A.5 Rank-Weighted Distributions of Objective Function Values of 2D Shape and Arithmetic Expression Datasets

Finally, to complement the rank-weighted distributions of objective function values of the ZINC dataset in Figure 2, we here also show the corresponding distributions for the 2D shape and arithmetic expression datasets used in Section 6 (Figure 7 and Figure 8).

Figure 7: Illustration of rank weighting (Equation (1)) on the shapes dataset (see Section 6) (similar to Figure 2).

Figure 8: Illustration of rank weighting (Equation (1)) on the arithmetic expression dataset (see Section 6) (similar to Figure 2).

# B Further Experimental Results

## B.1 Optimization Performance with More Weighted Retraining Parameters

Holding $r$ fixed at $r_{low}$, we vary $k$ from $k_{low}$ to $\infty$ (Figure 9) and vice versa (Figure 10). In general, performance increases monotonically as $k, r$ decrease, suggesting a continuous improvement from increasing weighting or retraining. The arithmetic expression task did not show this behaviour for retraining, which we attribute to the high degree of randomness in the optimization.

Figure 9: Top1 optimization performance of weighted retraining for different $k$ values with $r = r_{low}$.

Figure 10: Top1 optimization performance of weighted retraining for different $r$ values with $k = k_{\mathrm{low}}$ $r_{\mathrm{low}} = 5, r_{\mathrm{high}} = 50$ for the 2D shape area task; $r_{\mathrm{low}} = 50, r_{\mathrm{high}} = 100$ for the others

Figure 11: Top10 optimization performance of weighted retraining for all tasks (setup identical to Figure 4).

## B.2 Top10 and Top50 Optimization Results

Figure 11 and Figure 12 give the Top10 and Top50 scores for the experiment described in Section 6.2. These results are qualitatively similar to those in Figure 4, suggesting that our method finds many unique high-scoring points.

Figure 12: Top50 optimization performance of weighted retraining for all tasks (setup identical to Figure 4).

## B.3 Comparison of Chemical Design Results with Previous Papers

Table 1 compares the results attained in this paper with the results from previous papers that attempted the same task. Weighted retraining clearly beats the previous best methods, which were based on reinforcement learning, while simultaneously being more sample-efficient. Note that despite using the same pre-trained model as [28], we achieved better results by training our sparse Gaussian process on only a subset of data and clipping excessively low values in the training set, which allowed us to get significantly better results than they reported.

| Model | 1st | 2nd | 3rd | no. queries (source) |
|---|---|---|---|---|
| JT-VAE [28] | 5.30 | 4.93 | 4.49 | 2500 (paper[4]) |
| GCPN [75] | 7.98 | 7.85 | 7.80 | $\approx 10^6$ (email[5]) |
| MolDQN [77] | 11.84 | 11.84 | 11.82 | $\geq 5000$ (paper[6]) |
| ChemBO [36] | 18.39 | - | - | **100** (Table 3 of [36]) |
| **JT-VAE** (our Bayesian optimization) | 5.65 | 5.63 | 5.43 | 500 |
| **JT-VAE** ($k = 10^{-3}$, no retraining) | 5.95 | 5.75 | 5.72 | 500 |
| **JT-VAE** ($k = 10^{-3}$, retraining) | 21.20 | 15.34 | 15.34 | 500 |
| **JT-VAE** ($k = 10^{-3}$, retraining, *best result*) | **27.84** | **27.59** | **27.21** | **500** |

Table 1: Comparison of top 3 scores on chemical design task. Baseline results are copied from [77]. All our results state the *median* of 5 runs unless otherwise stated (judged by best result), each run being 500 epochs.

## B.4 Pictures of the Best Molecules Found by Weighted Retraining

Figure 13 illustrates some of the best molecules found with weighted retraining. Note that all the high-scoring molecules are extremely large. It has been reported previously that larger molecules achieve higher scores, thereby diminishing the value of this particular design task for RL algorithms [77]. However, the fact that these molecules were found with a generative model strongly highlights the ability of weighted retraining to find solutions outside of the original training distribution.

# C   Details on Experimental Setup

## C.1 Retraining Parameters

When retraining a model with frequency $r$, the model is optionally fine-tuned initially, then repeatedly fine-tuned on queries $r$, $2r$, $3r$, ... until the query budget is reached. All results use the rank-based weighting function defined in Equation (1) unless otherwise specified. We consider a budget of $B = 500$ function evaluations, which is double the budget used in [37, 28].

## C.2 Bayesian Optimization

For optimizing over the latent manifold, we follow previous work [37, 28] and use Bayesian optimization with a variational sparse Gaussian process (SGP) surrogate model [67] (with 500 inducing points) and the expected improvement acquisition function [30]. We re-implemented the outdated and inefficient `Theano`-based Bayesian optimization implementation of [37] (see `https://github.com/mkusner/grammarVAE`), which was also used by [28], using the popular and modern `Tensorflow 2.0`-based `GPflow` Gaussian process library [14] to benefit from GPU acceleration.

For computational efficiency, we fit the sparse Gaussian process (SGP) only on a subset of the data, consisting of the 2000 points with the highest objective function values, and 8000 randomly chosen

Figure 13: Some of the best molecules found using weighted retraining. Numbers indicate the score of each molecule.

points. This also has the effect of ensuring that the SGP properly fits the high-performing regions of the data. Disregarding computational efficiency, we nonetheless found that fitting on this data subset remarkably improved performance of the optimization, even using the baseline model (without weighted retraining).

## C.3 Evaluation Metrics

We report, as a function of the objective function evaluation $b = 1, \ldots, B$, the single best score obtained up until query $b$ (denoted as Top1), and the worst of the 10 and 50 best scores obtained up until evaluation query $b$ (denoted as Top10 and Top50, respectively). Since our goal is to synthesize entities with the desired properties that are both a) *syntactically valid* and b) *novel*, we discard any suggested data points which are either a) invalid or b) contained in the training data set (i.e., they are not counted towards the evaluation budget and thus not shown in any of the plots). For statistical significance, we always report the mean plus/minus one standard deviation across multiple random seeds.

## C.4 2D Shape Task Details

Figure 14 shows example images from our 2D squares dataset.

The convolutional VAE architecture may be found in our code. The decoder used an approximately mirror architecture to the encoder with transposed convolutions. Following general conventions, we use a standard normal prior $p(\mathbf{z}) = \mathcal{N}(0, 1)$ over the latent variables $\mathbf{z}$ and a Bernoulli likelihood $p(\mathbf{x}|\mathbf{z})$ to sample binary images. Our implementation used PyTorch [52] and PyTorch Lightning [18].

## C.5 Arithmetic Expression Fitting Task

Following [37], the dataset we use consists of randomly generated univariate arithmetic expressions from the following grammar:

```
S → S '+' T | S '*' T | S '/' T | T
T → '(' S ')' | 'sin(' S ')' | 'exp(' S ')'
T → 'v' | '1' | '2' | '3'
```

Figure 14: Sample images from our 2D squares dataset.

where S and T denote non-terminals and the symbol | separates the possible production rules generated from each non-terminal. Every string in the dataset was generated by applying at most 15 production rules, yielding arithmetic expressions such as sin(2), v/(3+1) and v/2 * exp(v)/sin(2*v), which are all considered to be functions of the variable v.

The objective function we use is defined as $f(\mathbf{x}) = -\log(1 + \text{MSE}(\mathbf{x}, \mathbf{x}^*))$, where $\text{MSE}(\mathbf{x}, \mathbf{x}^*)$ denotes the mean squared error between $\mathbf{x}$ and the target expression $\mathbf{x}^* = 1/3 * v * \sin(v*v)$, computed over 1000 evenly-spaced values of v in the interval between $-10$ and $+10$. We apply the logarithm function following [37] to avoid extremely large MSE values resulting from exponential functions in the generated arithmetic expressions. In contrast to [37], we negate the logarithm to arrive at a maximization problem (instead of a minimization problem), to be consistent with our problem formulation and the other experiments. The global maximum of this objective function is $f(\mathbf{x}) = 0$, achieved at $\mathbf{x} = \mathbf{x}^*$ (and $f(\mathbf{x}) < 0$ otherwise).

In contrast to the original dataset of size 100,000 used by [37], which *includes the target expression* and many other well-performing inputs (thus making the optimization problem easy in theory), we make the task more challenging by discarding the 50% of points with the highest scores, resulting in a dataset of size 50,000 with objective function value distribution shown in Figure 8.

Our implementation of the grammar VAE is based on the code from [37] provided at https://github.com/mkusner/grammarVAE, which we modified to use Tensorflow 2 [1] and python 3.

## C.6  Chemical Design Task

The precise scoring function for a chemical $\mathbf{x}$ is defined as:

$$\mathrm{score}(\mathbf{x}) = \max\left(\widehat{\log P(\mathbf{x})} - \widehat{\mathrm{SA}(\mathbf{x})} - \widehat{\mathrm{cycle}(\mathbf{x})}, \, -4\right)$$

where $\log P$, SA, and cycle are property functions, and the $\widehat{\phantom{x}}$ operation indicates standard normalization of the raw function output using the ZINC training set data (i.e. subtracting the mean of the training set, and dividing by the standard deviation). This is identical to the scoring function from references [37, 9, 28, 77, 75], except that we bound the score below by $-4$ to prevent points with highly-negative scores from substantially impacting the optimization procedure. Functionally, because this is a maximization task, this makes little difference to the scoring of the outcomes, but does substantially help the optimization.

Our code for the junction tree VAE is a modified version of the "fast jtnn" code from the authors of [28] (available at `https://github.com/wengong-jin/icml18-jtnn`). We adapted the code to be backward-compatible with their original pre-trained model, and to use pytorch lightning.

## C.7  Other Reproducibility Details

**Range of hyperparameters considered**  We originally considered $k$ values in the range $10^1, 10^0, \ldots, 10^{-5}$, and found that there was generally a regime where improvement was minimal, but below a certain $k$ value there was significant improvement (which is consistent with our theory). We chose $k = 10^{-3}$ as an intermediate value that consistently gave good performance across tasks. This value was chosen in advance of running our final experiments (i.e. we had preliminary but incomplete results with other $k$ values, then chose $k = 10^{-3}$, and then got our main results). The retraining frequency of 50 was chosen arbitrarily in advance of doing the experiments (specifically it was chosen because it would entail retraining 10 times in our 500 epochs of optimization). The hyperparameters for model design and learning were dictated by the papers whose models we chose, except for the convolutional neural network for the shape task, where we chose a generic architecture. For the baseline methods (i.e. DbAS, CEM-PI, FBVAE, and RWR), we identified the best hyperparameter settings using a grid search over a reasonable range. We used the following hyperparameter settings: a quantile parameter of 0.95 for DbAS, CEM-PI and FBVAE (for all benchmarks), a retrain frequency of 200 for all baselines and for all benchmarks, an exponential coefficient of $10^{-3}$ (for the shapes task) and $10^{-1}$ (for the expression and chemical design tasks) for RWR, and a noise variance of 10 (for the shapes task) and 0.1 (for the expression and chemical design tasks) for DbAS.

**Average run time for each result**  All experiments were performed using a single GPU. Runtime results are given in Table 2.

| Experiment | GPU hours per run |
| --- | --- |
| Shapes (model pre-training) | 0:20 |
| Shapes (optim., retraining) | 0:20 |
| Shapes (optim., no retraining) | 0:01 |
| Expressions (optim., retraining) | 3:15 |
| Expressions (optim., no retraining) | 1:45 |
| Chemical Design (optim., retraining) | 5:00 |
| Chemical Design (optim., no retraining) | 3:00 |

Table 2: Approximate runtimes of main experiments

**Computing infrastructure used**  All experiments were done using a single GPU (either NVIDIA P100, 2070 Ti, or 1080 Ti). In practice, a lot of the experiments were run on a high-performance computing cluster to allow multiple experiments to be run in parallel, although this was strictly for convenience: in principle, all experiments could be done on a single machine with one GPU.

## Footnotes

[4]These were the top results across 10 seeds, with 250 queries performed per seed.

[5]Obtained through email correspondence with the authors.

[6]The experimental section states that the model was trained for 5000 episodes, so at least 5000 samples were needed. It is unclear if any batching was used, which would make the number of samples greater.