[Reviews · NeurIPS 2020]

Review 1

Summary and Contributions: The authors extended optimization in the latent space of a generative model by weighting acquired samples and periodically retraining the generative model. Results on two synthetic and one real-world benchmark task show that weighting and retraining help to find the optimum of a function faster.

Strengths: The paper is clearly written.

Weaknesses: 1. Weighted retraining is not new. The cross-entropy method (De Boer et al., 2005; Neil et al., 2018) maximizes the expectation E_p(x)[f(x)] of the objective function f(x) when sampling from a policy p(x) by periodically retraining p(x) on the samples with the highest reward, e.g. those with a reward above a quantile cutoff (i.e. using a stepwise weighting function). Instantiations of the cross-entropy method include DbAs (Brooks et al) and FBGAN (Gupta et al). Reward weighted regression (RWR) (Hachiya et al) is another existing optimization technique that employs weighted retraining. Angermueller et al. (http://arxiv.org/abs/2006.03227) recently employed these techniques as baselines for high-dimensional discrete optimization. 2. The described rank-based weighting function is not new. See RankGAN (Lin et al. 2017) or LeakGAN (Guo et al. 2017) for an example. 3. The evaluation is missing important baselines such a DbAs, FBGAN, RWR, and model-based optimization. 4. Chemical design task: It is unclear how the optimization trajectory of ‘original’ was obtained. How were new data points sampled from JT-VAE? Why does the trajectory stop at 250? 5. In addition to JT-VAE, I would also like to see a comparison with GCPN (You et al) and reinforcement learning. 6. What do error bars represent? How often were experiments repeated with different random seeds?

Correctness: The methodology is correct.

Clarity: The paper is clearly written.

Relation to Prior Work: The related work section misses a broader discussion of cross-entropy optimization, reward weighted regression, and DbAs. Bayesian optimization has also been used for optimizing high-dimensional functions (ChemBO, http://arxiv.org/abs/2006.03227).

Reproducibility: Yes

Additional Feedback: How do you explain the sudden increase of ‘weight; retrain’ in Figure 4 (right) and the good performance of weighting without retraining in Figure 4 (left)?


Review 2

Summary and Contributions: The paper proposes a method to improve the general performance and sample efficiency for sample latent space optimization. Here the value of an expensive to evaluate black-box objective function is seeked maximized using as few evaluations of the black box functions as possible. The paper follows previous literature and does the optimization in a (learnable) latent space where the optimization is presumably easier. The main contributions of the paper to first identify that the structure of the latent space is normally decoupled from the black-box function and then propose two simple solutions to alleviate that problem: 1) reweight the training data according to the objective function and secondly to retrain the (DGM) latent space model. # Author rebuttal: I thank the authors for their rebuttal and would like to acknowledge that I read and considered their answers!

Strengths: Pros: The paper tackles an important and often encountered problem of optimization of a general black box function. The authors clearly identify a problem with the current previous literature and proposes a simple, almost trivial (in the good way), solution to the problem The experimental results support that the proposed solution works well and are both more sample efficient as well as achieving better ‘objective function’ results.

Weaknesses: Cons: In general I found the method section ok, however some important parts are missing and need to be addressed. “Fit objective model h” (pseudo algo line 6) What is h and how is it fitted. You mention a gaussian process for the Zinc dataset - why is that model appropriate and how well does it actually fit the true objective function? “suggest new latent z ̃ based on h” (pseudo algo line 6) How do you find new latent space samples? Gradient descent on h? Some of this information can likely be found in the refs or in the appendix however this information (in my opinion) really needs to be explained and self-contained in the main paper It would strengthen the paper a lot if one more real world example were included in the experimental results (currently two toy tasks, one real world dataset). I’m missing some analysis on the performance of the DGM models - .e.g how do the likelihood (ELBO) change due to the weighted training. Especially it could be interesting to see an experiment showing that the weighting+training actually decreases the likelihood of data points with low objective function values. I found the discussion of the exact weighting implementation unnecessary confusing 143-146L. The chosen implementation where the data points are repeated according to the inverse weight seems to be a complex and approximate solution to simply sampling according to the (normalized weights) with replacement? (please confirm or correct me if I’m wrong and rewrite that section as appropriate)

Correctness: Claims seems correct and sound

Clarity: In general the paper is well written and easy to follow

Relation to Prior Work: Nothing to add

Reproducibility: Yes

Additional Feedback: Wrt to the weaknesses i would like the authors to address the following Rewrite parts of method section wrt to surrogate objective function h and how the optimization in the latent space is done do address the comments above - This is of high importance to the quality of the paper Add analysis of the quality of the DGM models and how the weighting+retraining shifts the probability density. Optionally add another real world dataset or discuss/address why the chosen datasets are sufficient If the above are addressed I would be happy to bump my score


Review 3

Summary and Contributions: This paper proposes improvements to current latent space optimization (LSO) methods which train a mapping from a lower dimensional latent space to the true input space of the objective function allowing for the more efficient optimization of the latent space. The authors conjecture the way current LSO methods train and use the latent space mapping severely limits their effectiveness. They explain that this is due, at least in part, to training the deep generative model (DGM) on a large proportion of low performing examples as well as the lack of retraining to leverage new samples. They formulate some intuitive arguments for why this might be true by discussing the properties of what they call the "feasible region" representing the region in latent space containing most of the probability mass. To alleviate these issues, the authors propose to regularly retrain on the weighted dataset. By adjusting the weights of individual data points based on their objective value, they show that they can leverage new data points, i.e., function evaluations, while avoiding catastrophic forgetting. Additionally, they argue that the weighted dataset allows the DGM to capture a large and higher-valued portion of the input space. Some empirical results are provided on a 2D shape area maximization task, an arithmetic expression fitting task, and a chemical design task.

Strengths: This work seems well motivated and sound. While retraining and importance weighting aren't novel ideas, I consider the main contributions of this work to be come from identifying and isolating issues with how LSO is currently used. These types of contributions are relevant of the NeurIPS community and can have considerable impact. That being said, I am not an expert in this field so I don't consider myself a good judge of the significance of this work or how this works fits into the existing literature. The author's conjectures are intuitive and sound. The empirical evaluation supports the author's conjectures giving them some credence.

Weaknesses: There are no obvious weaknesses along those axes that I can think off given my limited knowledge of this field.

Correctness: The claims and method seem correct. One major (but easily fixed) issue with figure 4 is the use of the standard error to display bounds on a sample mean computed with 3 samples. There is no reasonable expectation that this sample mean is normally distributed making the standard error a meaningless measure of uncertainty. For such low number of seeds, shading with the min/max or even plotting all three curves is much more meaningful. Alternatively, the empirical standard deviation could be used but, while not erroneous to use in this context like the standard error, it is arguably not very meaningful when summarizing 3 samples.

Clarity: The paper is well written and very clear. I enjoyed reading it.

Relation to Prior Work: This paper exhaustively discusses related work. While it would be unlikely for me to catch any omissions, I feel like I have a good understanding about the context surrounding this work.

Reproducibility: Yes

Additional Feedback: - Line 257, Given the training procedure is random, wouldn't their always be randomness to the optimization procedure from the trained mapping? Have the authors examined how much variance is introduced by their retraining approach? - The meaning of the shaded areas and solid lines of figure 4-right is not mentioned. I assumed it was the same as for the left. If that is the case, the same comments apply.


Review 4

Summary and Contributions: The paper proposes a method to increase the sample efficiency of latent space optimization (LSO) over structured domains such as molecules. The two key ideas are to place greater weight on higher-scoring data points (with respect to some property) during training and repeatedly retrain the model with new data points queried during property optimization. The paper identifies two limitations of the current LSO paradigm, proposes the two-part method above, and tests it on two toy tasks (binary shape area optimization and arithmetic expression fitting) as well as a molecule search task.

Strengths: Good empirical evaluation: The empirical results appear strong. There is little reason to expect that a general latent space learned by a generative model should necessarily be best-suited for property optimization. The evaluation on three tasks and associated ablation studies (with and without weighting and retraining), show that both components of the method contribute to the observed performance improvement relative to baselines. Simple method: The proposed method is quite simple and should be easy for other researchers to try in their pipelines. Relevance: I believe this work is relevant to the NeurIPS community. Optimizing structured/discrete objects via learned latent representations is a standard methodological route now, and this paper attempts to alleviate some of its current limitations.

Weaknesses: Discussion: The paper could have used a bit more analysis of the results with respect to the intuitive arguments presented in sections 3 and 4. For example, for the molecular search task, a molecule is found with 22.55 logP score. Where in the latent space would this molecule have been located for the generic baseline (i.e., no weighting or retraining). Is it actually outside the feasible region? If not, perhaps something else changes about the latent space that makes it more amenable to optimization? Related work: This is minor, but the related work section could be improved slightly with respect to Bayesian optimization (see related work section below).

Correctness: The method and evaluation appear to be sound.

Clarity: The paper is well-written.

Relation to Prior Work: Yes overall. I believe the Bayesian optimization (BO) section of the related work could be adjusted to touch on some of the BO work that has occurred in the chemical design space. The authors make the claim that no efficient BO methods exist for this domain, but I'm not sure that's true. Several of the works cited in the paper do indeed incorporate BO for chemical design, and a discussion of their claimed limitations would be worthwhile.

Reproducibility: Yes

Additional Feedback: Can the authors comment on why they think the reweight only baseline for the arithmetic expression task recovered a substantial portion of the weight+retrain performance? This is not observed for the other two tasks. I think some analysis of the kind mentioned in the weaknesses section would be interesting to see. The method assumes one already has access to the ground truth property scores for the training set examples. I'm curious how the authors would go about adapting the method for cases where only a subset of the training set is labeled. Added after the authors provided their rebuttal: Thanks to the authors for addressing most of the questions. The authors provided a short analysis of the log prob of the best point as retrainings take place. I think this type of analysis is crucial to support the many intuitive arguments presented in the paper. I still think the claim that no BO methods can handle the structured domains explored in the paper efficiently is a bit tenuous (e.g., JT-VAE has a BO demonstration). Even if the presented method outperforms others, this claim is too strong. Overall I believe the paper meets the acceptance threshold. However, I do agree with some of the points raised by R1 regarding further comparisons to specific baselines being helpful for a more rigorous evaluation, so ideally the authors can address that.

[Author Response · NeurIPS 2020]

**Summary.** We thank the reviewers for their detailed and insightful comments. We are pleased that the reviewers highlighted the importance of our motivation and relevance to the NeurIPS community (R2,R3,R4), our simple and easy-to-implement solution (R2,R4), and our strong empirical results (R2,R3,R4). We are also glad that they unanimously appreciated our clarity of writing, the soundness of our claims, reproducibility of our results, and correctness of our methodology (R1,R2,R3,R4). We are happy that, as a result, the overall sentiment was positive, with three reviewers (R2,R3,R4) recommending acceptance. R1 was mainly concerned with the novelty of our work, and suggested that we discuss additional literature and consider more baselines. We address the reviewers' concerns and questions below.

**Novelty of contributions.** R1 was concerned that weighted retraining and rank-based weighting functions are not novel. We agree, but *do not claim these ideas to be novel contributions of our paper*. We view our core contributions to be: 1) identifying critical failure modes of LSO, 2) addressing those issues by weighted retraining, and 3) showing that this "simple, almost trivial (in the good way), solution" (R2) yields *substantial* improvements on important practical problems (l. 44–50 of our paper). We like R3's comment: "While retraining and importance weighting aren't novel ideas, I consider the main contributions of this work to come from identifying and isolating issues with how LSO is currently used. These types of contributions are relevant of the NeurIPS community and can have considerable impact."

**Comparison to cross-entropy (CE) method.** R1 has pointed out that CE has similarities to our proposed method, but is not discussed in our paper. We acknowledge this, and want to highlight two main differences between the methods. Firstly, *standard CE produces only binary weights* (Boer et al), which amounts to simply adding or removing points from the training set. This is sub-optimal for reasons discussed in our paper (lines 123–9, 162–4), and consequently *we consider a strictly more general form of weighting*. Secondly, *CE has no intrinsic optimization component*. High-performing points are found only by repeatedly sampling from the generative model and evaluating target function $f$. By contrast, our method *explicitly selects high-performing points* using Bayesian optimization. The necessity of repeated sampling in CE makes it only suitable in cases where evaluating $f$ is cheap, which is *not* the problem scenario that we are considering, and therefore we did not consider it to be a useful baseline. However, we agree that the similarities to CE are enough to warrant discussion in

Figure 1: More baselines (left); $\log p$ over time (right)

the camera-ready version. For completeness, at the request of R1, we ran some CE methods used in Angermueller et al. and Brookes et al. on our easiest task (2D shapes), where they *significantly underperform* our method (Fig. 1). Note that we did extensive hyperparameter tuning of the baselines, and plot only the best result across three random seeds.

**Comparison to model-based/Bayesian optimization for chemistry** (R1, R4). Model-based optimization methods in input space (e.g. ChemBO) require highly engineered search strategies infused with a lot of domain knowledge (e.g. molecular distance measures and synthesis graphs) to perform well. They are thus designed for a specific problem class (e.g. chemical design) and not applicable to other problems (e.g. the shape and expression tasks we consider). In contrast, our method can be applied to any problem without domain knowledge. Despite this generality, even for chemical design, our method is competitive with ChemBO (a SOTA model-based method), achieving a final penalized logP score of 22.55, which is even higher than the final score of 18.39 reported in the ChemBO paper (see their Table 3).

**Other (mostly minor) questions and concerns raised by the reviewers.**

- R2 & R4 suggested that we examine the trajectory of high-scoring points in latent space. Fig. 1 shows $\log p(\mathbf{x})$ for the overall best point $\mathbf{x}$ (with score 22.55) with 3 seeds. $\log p(\mathbf{x})$ was estimated with importance sampling using 1000 samples from the encoder. The increase supports our conjecture that such points move into the feasible region.
- *Comparison with RL* (R1). Table 1 in Appendix B.5 favourably compares our method with GCPN and MolDQN.
- *"Original" line in Fig. 4 (right)* (R1). These are the original results from Jin et al, highlighting our improvement.
- *Sudden increase of 'weight; retrain' in Fig 4* (R1). The retraining every 50 epochs incorporates new information into the latent space, causing performance to increase (l. 284-87 in our paper).
- *Why is 'weight; retrain' much better than 'weight; no retrain' on all tasks except in Fig. 4 (left)?* (R1,R4). 'weight; no retrain' already comes close to the optimal score of 0, leaving less room for further improvements by retraining. In contrast, in Fig. 3 (left) the best score is much harder to achieve, and in Fig. 4 (right) it is even unbounded.
- *Error Bars*. Error bars represent standard error over 3 seeds (R1). This accounts for the randomness in the optimization procedure (R3). Negligible variation was observed for the shapes task (R3). We agree that standard error is not the best metric in this case, and will replace it with empirical standard deviation (R3).
- *What predictive model do you use?* (R2). Our method does not require a specific implementation for $h$. In our experiments we used Gaussian processes, which are universal function approximators with principled uncertainty estimates. Their use is standard practice in the Bayesian optimization literature. We will clarify this in the main text.
- *"How do you find new latent space samples?"* (R2). We optimize the expected improvement function using L-BFGS.

[Meta-Review · NeurIPS 2020]

This paper had 4 qualified reviewers, 3 of whom recommended acceptance and one who gave a 4 (updated from a 3 post-rebuttal). I think some of the complaints raised by the low review are technically correct, but I also don't feel that they are super-relevant to evaluating the scientific significance of this work. (i.e. I think the numerical score they gave was too low given the text of their review). Given all of that, I am recommending acceptance for this paper.